# Enabling Non-Technical Users to Create Knowledge Views

Alex Randles[1,*], Lucy McKenna[1], Lynn Kilgallon[2], Peter Crooks[2], Declan O'Sullivan[1], Jane Ohlmeyer[2], Claire McNulty[2], Felix Vanden Borre[2] and Daniel Patterson[2]

[1]ADAPT Centre, School of Computer Science and Statistics, Trinity College Dublin, Dublin, Ireland
[2]Department of History, Trinity College Dublin, Dublin, Ireland

### Abstract

The Virtual Record Treasury of Ireland (VRTI) is a project focused on the digital reconstruction of archival records damaged during a fire in the 1922 Irish Civil War. The initiative involves extensive interdisciplinary collaboration between historians and computer scientists. Complementing these reconstructed records is the Knowledge Graph for Irish History (KGIH), which provides a structured representation of historical entities, relationships and associated resources. The VRTI-KG Explorer is an interface that enables both researchers and the general public to search, explore and visualise data contained within the KGIH. During the course of the project, a key requirement emerged: historians needed a way to group and visualise entities within the KG without requiring expertise in semantic technologies or graph query languages. Historians wanted to curate and present collections of people connected through particular themes, events, or contexts, but lacked the technical background necessary to interact directly with graph-based data systems. To address this challenge, this work describes an approach for enabling intuitive and user-friendly visual exploration and organisation of entities within historical KGs. The "KG Views" described in this paper were developed to support historians in curating, organising and presenting entities and visual resources from the KGIH through the VRTI-KG Explorer in a more accessible and meaningful way.

### Keywords

Knowledge Graph Views, User Interface, Digital Humanities

## 1. Introduction

The Virtual Record Treasury of Ireland (VRTI)[1] [1] is an ambitious digital humanities project, which aims to virtually reconstruct archival records that were destroyed during the 1922 Irish Civil War. During the conflict, a fire in the western block of the Four Courts complex in Dublin destroyed centuries of historical records, resulting in one of the most significant archival losses in Irish history. Through extensive digitisation and reconstruction efforts, the VRTI now provides free access to more than 350,000 records and over 250 million searchable words documenting seven centuries of Irish history.

To enrich access to these historical resources, the Knowledge Graph for Irish History (KGIH)[2] [2] was developed to represent information about notable Irish people, places, organisations, events and other entities in a structured graph format. The KGIH contains almost three million interconnected triples and is linked to external knowledge bases including Wikidata and DBpedia, enabling richer contextualisation and interoperability with the wider Linked Open Data ecosystem.

To facilitate exploration of the KG, the KG Explorer platform [3, 4] was developed as a user-facing interface for browsing and discovering relationships within the KGIH. Since its launch, the platform has been used by more than 10,000 users across over 90 countries. As part of the integration between the VRTI and KGIH, more than 4,000 archival records have been linked to entities represented in the KG. These connections enable users to discover previously hidden relationships between historical records, individuals, locations, organisations and events. Archival records are associated with KG resources to indicate the involvement of an entity within a particular historical document, creating new pathways for exploration and research.

*Second International Workshop on Users and Knowledge Graphs (UKG), co-located with SEMANTiCS'26: International Conference*
*Corresponding author.

✉ [alex.randles@adaptcentre.ie](mailto:alex.randles@adaptcentre.ie) (A. Randles)

[1]https://virtualtreasury.ie/
[2]https://kg.virtualtreasury.ie/

As adoption of the platform has grown, historians identified a need to organise and present thematic collections of related entities without requiring expertise in semantic technologies or graph query languages. To address this challenge, this paper presents KG Views, a framework that enables the creation, management and visual exploration of curated collections of entities and associated resources within the KGIH, through an intuitive interface integrated into the existing VRTI-KG Explorer. Since its deployment within the VRTI ecosystem, KG Views has been used to create a range of thematic collections, improving the accessibility, discoverability and presentation of graph-based data.

This paper is structured as follows: Section 2 presents the design requirements derived from workshops with the VRTI historians. Section 3 describes the design and implementation of the KG views. Section 4 describes views that have been created by historians. Section 5 describes initial feedback from users of the editor. Section 6 discusses approaches designed to curate and managed scholarly KGs. Section 7 concludes the paper and outlines future work.

## 2. Design Requirements

The following design requirements emerged from workshops with historians involved in the VRTI, aimed at identifying their needs when creating KG views. An initial set of requirements was defined and subsequently refined through iterative prototype testing.

- Intuitive interaction through a user-friendly interface
- Ability to use the system independently, without assistance from computer scientists
- Visualisation of related geospatial information on an interactive map
- Support for adding and highlighting featured entities

These requirements were used to guide the development of integrated features to facilitate the creation and visualisation of KG views in the explorer.

## 3. Implementation of KG Views

The KG views were integrated into the existing VRTI-KG explorer [3] through the implementation of a view editor interface, which allows historians to add and manage information associated with individual views.

### 3.1. Views Editor

This section describes the final version of the implementation of the views editor. The editor was iteratively refined based on the feedback from the historians and until the functionality satisfied the design requirements, outlined in Section 2. The inputs in the editor include validation to ensure that values entered are valid for the attribute. The editor was created using a combination of technologies, shown below:

- **Flask**: A lightweight Python web framework used to provide the application's backend functionality, routing and integration with the underlying data model.
- **Bootstrap**: A front-end framework used to implement the view editor's user interface, providing responsive layouts, form components and consistent styling for configuring and managing view definitions.
- **JSON**: Used as the data interchange format for the view configurations and metadata between the client and server.

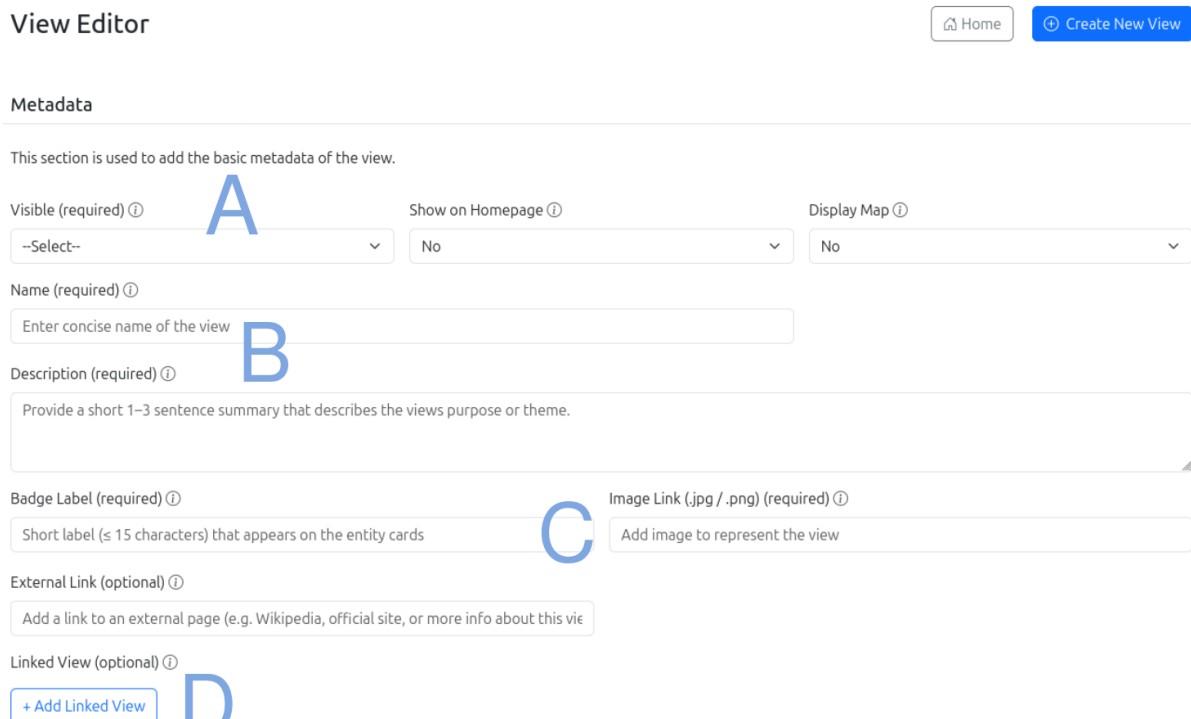

**Figure 1:** Basic metadata in views editor

## 3.2. Metadata of Views

Figure 1 presents a screenshot of section of the views editor to populate the metadata associated with a view.

Users can set the visibility (A) of the view to either appear on the homepage or be visible only within sub-views. A map can be dynamically generated for the view by retrieving geospatial information associated with it. Users can then add the name (B), description and image associated with the view. An HTTP request is executed to validate that the image URL (C) is valid. In addition, a badge label is used as a tag to indicate which entities in the KG are associated with this resource. Finally, related views (D) can be selected from a drop-down menu to indicate that they are linked.

## 3.3. Associating Entities with a View

Figure 2 presents a screenshot of the section of the editor used to associated entities with the view.

All entities in the KGIH are associated with views by providing an existing KG Explorer URL (A) containing the desired search filters. For example, to create a view displaying all people associated with documents from the 1641 depositions collection, which contains witness testimonies relating to the Irish Rebellion of 1641, a user can provide the following URL generated through the advanced search functionality of the explorer:

**https://kg.virtualtreasury.ie/search-results?People:Document=https://1641.tcd.ie/**

These URL query parameters are parsed and translated into SPARQL filter expressions, which are then used to generate a query that retrieves all resources associated with the selected view.

FILTER (?Document = <https://1641.tcd.ie/>)

In addition, users can provide a list of KG URIs (B) to be associated with the view. These URIs can be used alongside the SPARQL filters to further refine or constrain the set of resources returned by the query. Users may enter URIs directly into the text area or upload a spreadsheet (C) containing a list of

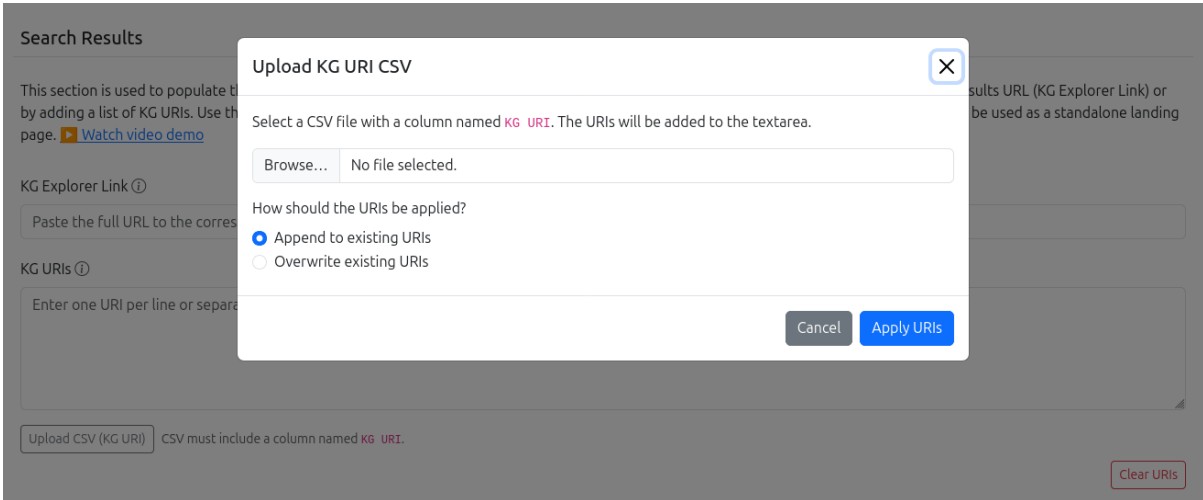

**Figure 2:** Adding associated entities in editor

**Figure 3:** Uploading associated entities in editor

URIs. Figure 2 presents the URI upload interface. The uploaded URIs can either be appended to the existing list or used to overwrite it entirely.

### 3.4. Featured Entities

The views editor allows users to add notable entities from the view in order to highlight them on the view's homepage. Figure 4 presents a screenshot of the section of the editor used to populate featured entities.

Individual featured entities (A) can be added by providing pairs of KG URIs and optional image URLs. In addition, users can add featured entities in bulk by uploading a spreadsheet (B) containing columns for KG URIs and image URLs. User input is validated to ensure that the KG URIs are in a valid format and that the image URLs resolve successfully.

## 4. Use Cases

VOICES [5, 6] is a European Research Council project with the aim of uncovering lived experiences of women in early modern Ireland (c.1550-c.1700). This section describes the VOICES view [3] which was

[3]https://kg.virtualtreasury.ie/views-homepage/VOICES

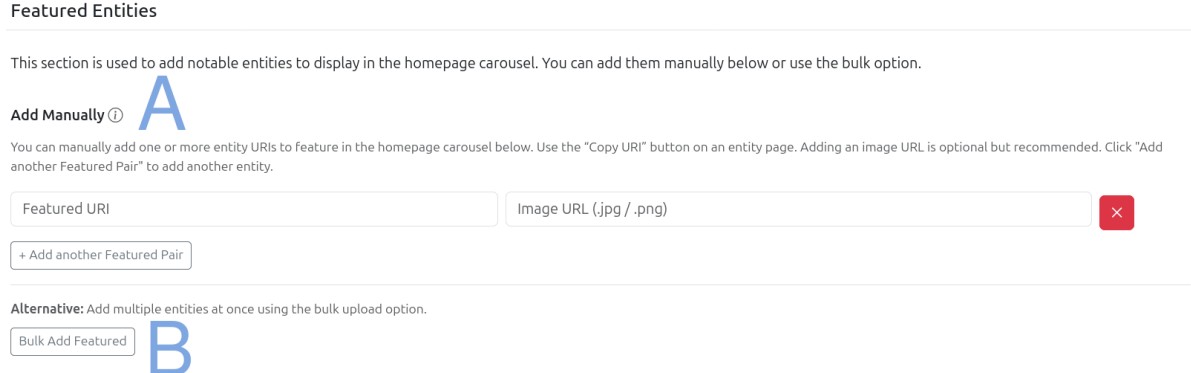

**Featured Entities**

This section is used to add notable entities to display in the homepage carousel. You can add them manually below or use the bulk option.

**Add Manually** ⓘ A

You can manually add one or more entity URIs to feature in the homepage carousel below. Use the "Copy URI" button on an entity page. Adding an image URL is optional but recommended. Click "Add another Featured Pair" to add another entity.

| Featured URI | Image URL (.jpg / .png) | ✕ |

[ + Add another Featured Pair ]

**Alternative:** Add multiple entities at once using the bulk upload option.

[ Bulk Add Featured ] B

**Figure 4:** Adding featured entities in editor

created to provide a dedicated interface for exploring and interacting with the VOICES entities in the KGIH.

The configuration of the view is defined through the KG Views editor, where user-defined settings are stored as a JSON object and dynamically rendered within the Flask web application. Listing 1 presents an extract from the configuration used to define the VOICES view.

Listing 1: Extract from the VOICES project configuration

```
 1  "VOICES": {
 2      "visible": true,
 3      "show_homepage": true,
 4      "display_map": true,
 5      "description": "The ambitious VOICES project ..",
 6      "badge_label": "VOICES",
 7      "image_link": "https://voices.sirv.com/artworks/...",
 8      "external_link": "https://voicesproject.ie/",
 9      "explorer_link": "https://kg.virtualtreasury.ie/search-results?...",
10
11      "related_uris": [
12        "https://kg.virtualtreasury.ie/place/present-day/town/Drogheda/v1xmx97"
13      ],
14
15      "featured": [
16        {
17          "uri": "https://kg.virtualtreasury.ie/person/Taylor_Mary_c17/v1bh2t2",
18          "image": "https://voices.sirv.com/artworks/archive-ugent-be.jpg"
19        }
20      ],
21
22      "linked_views": [
23        "VOICES Archive and Collection",
24        "VOICES Geography",
25        "VOICES Notable Women"
26      ]
27  }
```

The views configuration schema consists of a collection of presentation, navigation and discovery metadata. The `visible` field indicates whether the project is publicly displayed, while `show_homepage` determines whether it is featured on the homepage and `display_map` specifies whether it appears in map-based visualisations. A textual summary of the project's aims and scope is provided through `description` and a short identifier used in the user interface is defined by `badge_label`. Links to project resources are supplied through `image_link`, which references a representative project image,

external_link, which points to the project's external website. The explorer_link, which provides a link to the search results page in the explorer that contains desired search filters to associate with the view. The related_uris field contains a list of specific URIs associated with the view. Notable entities associated with the project are specified in the featured array, where each entry contains a URI and an optional image. Finally, linked_views defines a set of related views available for exploring related content.

Figure 5 presents the main entry point for accessing the view through the homepage of the explorer. Although implemented as KG Views, these are presented to users as **Showcases** in the interface to provide a clearer, more intuitive experience for non-technical users.

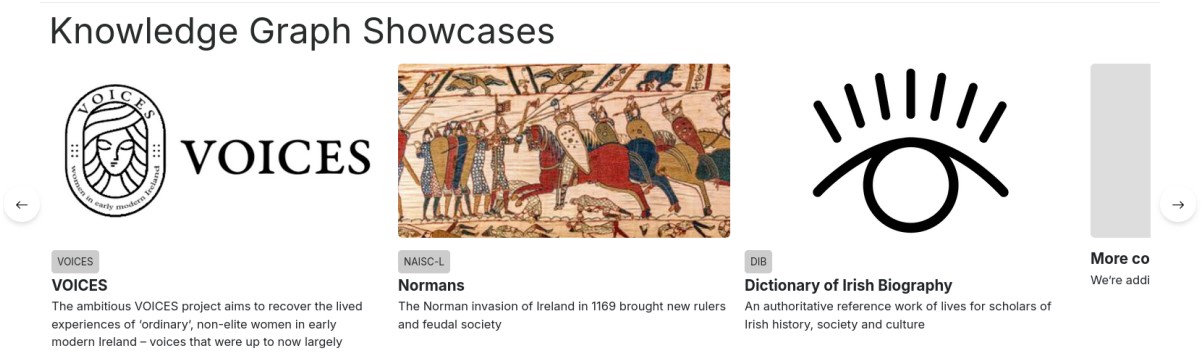

**Figure 5:** Carousel of views available on explorer

The VOICES view can be accessed by clicking the corresponding carousel item on the homepage of the KG Explorer. Then, the users are redirected to the homepage of the VOICES view (Figure 6).

The homepage displays the metadata configured through the editor, including the view's name, description and image. A search bar (A) enables users to enter search terms to discover relevant entities within the view. In addition, two links (B) are provided: one allows users to browse all entities contained in the view using the explorer, while the other directs users to external resources for further information about the view. Figure 7 presents the featured people carousel for the VOICES view.

This component highlights notable individuals associated with the view and provides direct navigation to relevant content about them. Figure 8 presents the carousel of sub-views associated to the VOICES view.

These related views allow users to explore specific aspects of the main VOICES view in greater depth. For example, the Geography view highlights notable locations associated with individuals, while the Archive and Collection views group people according to their source documents. The Notable Women view showcases prominent women featured within the view. Figure 9 presents the geospatial map automatically generated from the entities contained within the VOICES view. The map visualises the geographic information associated with these entities, enabling users to explore their spatial distribution and identify relevant locations represented in the KG. Figure 10 presents the search results page for the VOICES view.

This page indicates to users (A) that they are searching over the entities contained within the VOICES view. Each entity is displayed with an image and a selection of metadata retrieved from the KG. Users can switch between different entity categories, such as people and places, using the available navigation controls. In addition, faceted filters enable users to refine the search results according to specific attributes and characteristics represented in the KG. Users are redirected to the corresponding entity card when they select a search result item. Figure 11 presents an example of an entity card for a person.

The entity card provides a detailed overview of the selected individual, including biographical information, associated entities and relationships extracted from the KG. Users can further explore these connections by navigating to related entities, facilitating the discovery of contextual information and supporting exploratory search within the collection. As can be seen, the entity card contains a badge indicating that the resource is a member of the VOICES view. This badge acts as an interactive link,

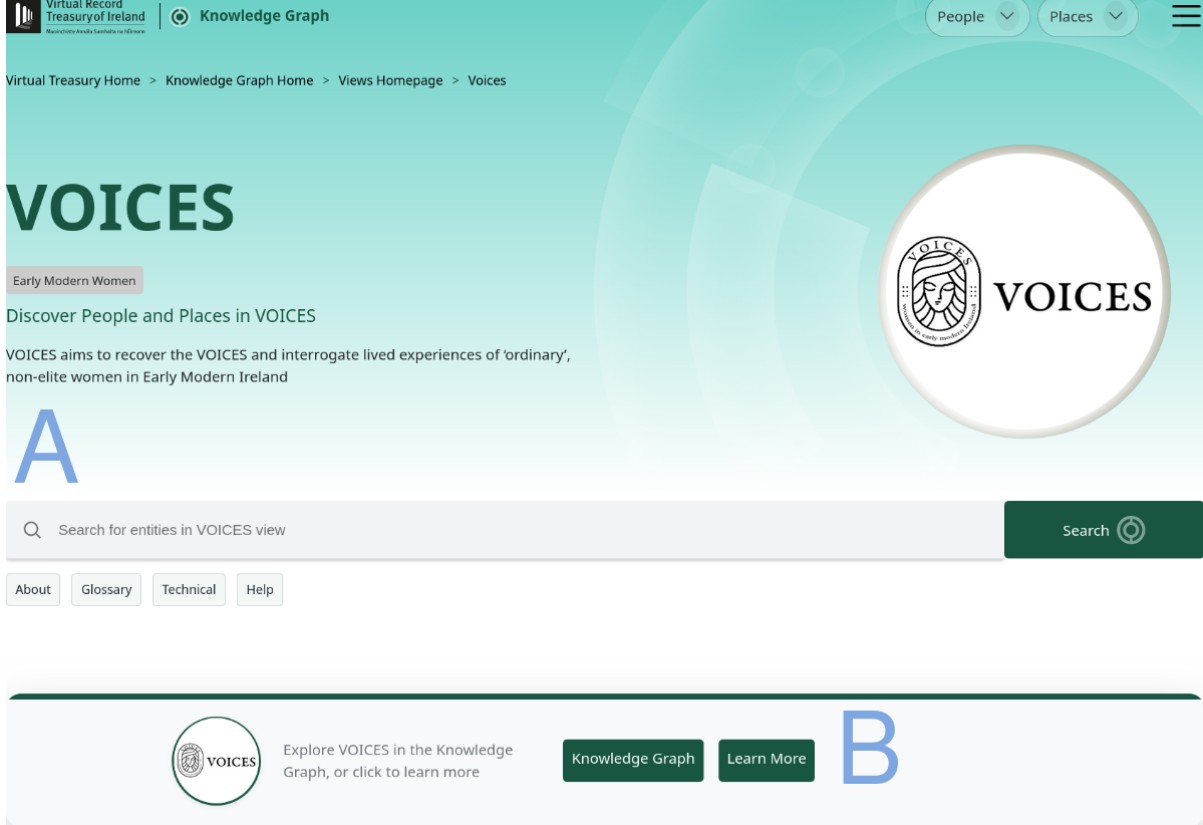

**Figure 6:** Homepage for VOICES view

## Featured People

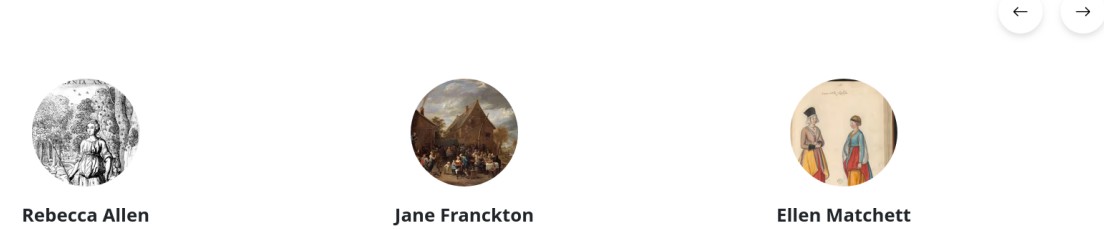

Rebecca Allen  Jane Franckton  Ellen Matchett

**Figure 7:** Featured people in VOICES view

allowing users to navigate directly to the corresponding view and explore additional related entities and resources associated with it.

Other notable views which have been created by the historians, so far include:

- **Dictionary of Irish Biography (DIB):** This view[4] contains biographical information on notable individuals associated with Ireland, derived from the authoritative Dictionary of Irish Biography[5]. The resource documents both Irish-born figures and individuals from elsewhere whose lives and careers made significant contributions to Irish society, history and culture.
- **Normans:** This view[6] contains information on Norman individuals and their historical activities, with a particular focus on their role in Ireland. Originating from the descendants of Norse settlers

---

[4]https://kg.virtualtreasury.ie/views-homepage/DictionaryofIrishBiography
[5]https://www.dib.ie/
[6]https://kg.virtualtreasury.ie/views-homepage/NormandyandIreland

## Related Showcases

**Geography**
**VOICES Geography**
Explore geographical locations associated with the VOICES project

**Archives**
**VOICES Archive and Collection**
Explore archival collections and documents from the VOICES project including 1641 Depositions, Freewomen records, and Wills

**Notable Women**
**VOICES Notable Women**
Notable women from the VOICES project including Janes and the Barnewall family

**More co**
We're addi

**Figure 8:** Views related to VOICES view

## Featured Maps

View full screen

Discover Locations for Births, Deaths and Residences

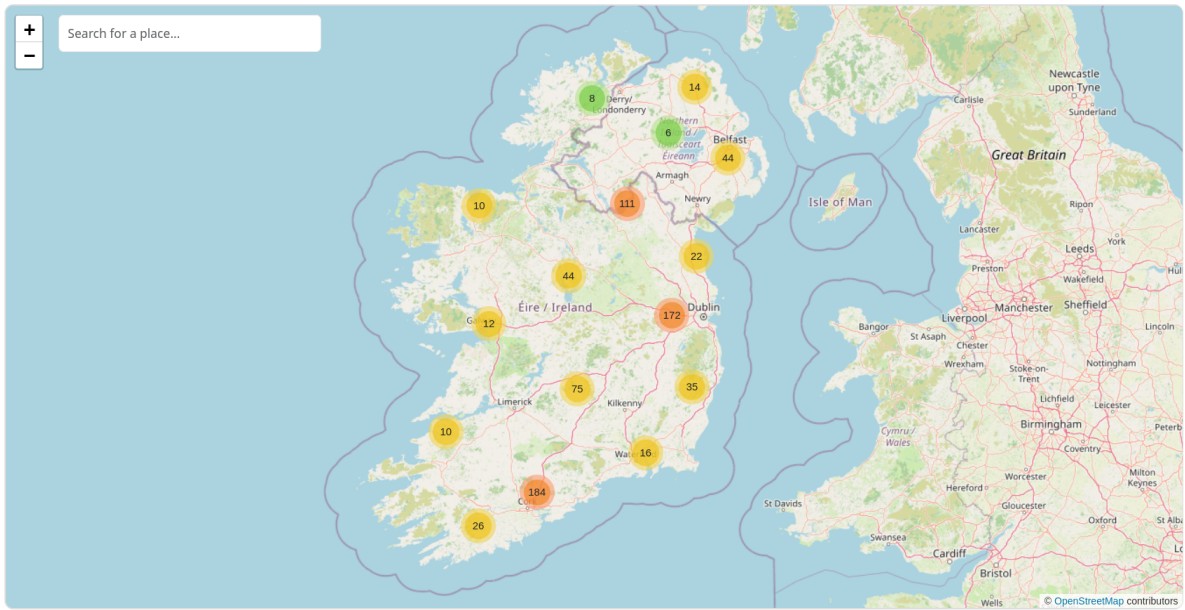

**Figure 9:** Geospatial map of VOICES view

and native Frankish populations in Normandy, the Normans became one of the most influential groups in medieval Europe. The dataset includes people and places associated with the Norman invasion of Ireland, which began in 1169 when Norman forces landed.

## 5. Initial Feedback from Users

Historians involved in the VRTI and VOICES projects have actively used the editor to create and manage views. The views functionality was only finalised and fully tested immediately prior to the July 2026 VRTI launch. Consequently, a formal user study evaluating either historians' experiences of generating views or public users' experiences of interacting with views has not yet been conducted. Instead, the initial versions of the views editor were iteratively refined based on informal feedback gathered from domain experts throughout the development process.

Key insights identified during this iterative refinement process are summarised below:

- **Image URL validation:** Users often provided webpage links instead of direct image URLs,

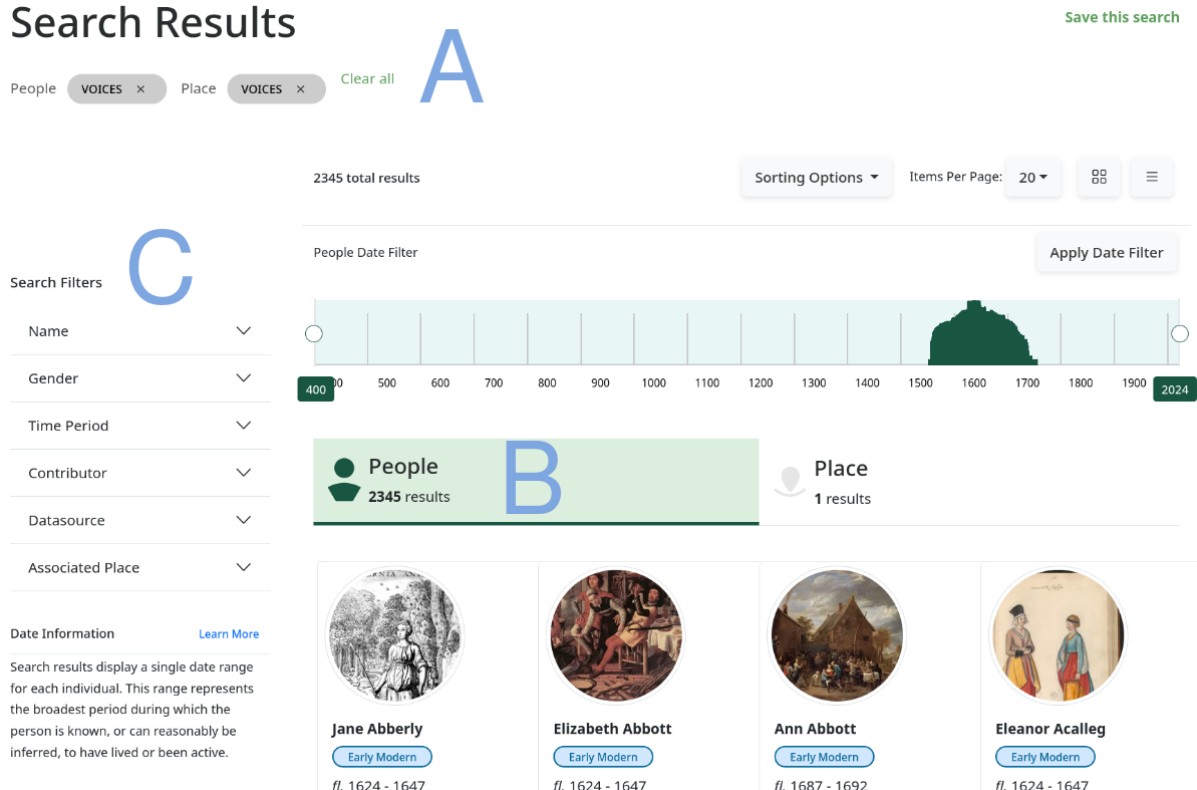

**Figure 10:** Search results of VOICES view

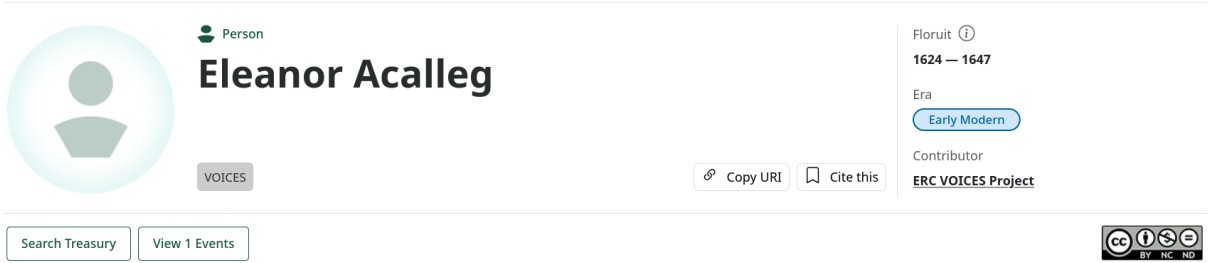

**Figure 11:** Sample person linked to VOICES view

making validation necessary to prevent display errors.

- **Linking between views:** Enabling links between related views was important for improving navigation and contextual exploration.
- **Reducing technical complexity:** Historians preferred interfaces that avoided exposing SPARQL queries or semantic web terminology, highlighting the importance of abstraction and simplicity in the editor design.
- **Support for incremental curation:** Historians often created views iteratively over time, requiring the ability to easily edit, expand and reorganise existing views.

It is hoped these key insights can be used by researchers to inform the design of similar interfaces that allow non-technical users to group together and visual entities in a KG.

# 6. Related Work

This section discusses approaches used to curate, manage and visualise digital humanities KGs.

WissKI [7] is an interface tool designed for managing scholarly KGs. It is integrated with the existing Drupal web content management system and leverages scholarly ontologies, such as CIDOC CRM, to generate the underlying data model used to support editing and data management. The Pathbuilder in WissKI enables the configuration of the platform for specific KGs. The platform requires users to possess a relatively advanced understanding of RDF ontology terms and semantic technologies, which can limit accessibility for non-technical users. While WissKI primarily focuses on data modelling and management, the approach proposed in this work extends beyond traditional KG editing by supporting user-friendly visual exploration and narrative organisation of entities.

The Open Research Knowledge Graph (ORKG) [8] is an infrastructure for the acquisition, curation, publication and processing of semantic scholarly knowledge. It supports collaborative KG construction and semantic scholarly communication through crowdsourcing and text mining approaches. While ORKG focuses on structuring and communicating scholarly research knowledge, this work focuses on enabling non-technical historians to visually explore, organise and group entities within the KGIH. The proposed KG views provide a more accessible and intuitive way for historians to interact with historical entities and visual resources without requiring expertise in semantic technologies or graph queries.

This work [9] discusses the evolution of semantic portals for cultural heritage across three generations: from linked data search and browsing, to analytical tools for digital humanities research and finally towards AI-driven systems capable of automated knowledge discovery and explainable reasoning. In addition, this work explores how semantic technologies support humanities research and highlights usability barriers in semantic systems. While this work focuses on AI-supported semantic portals and computational analysis, the KG views approach focuses on enabling non-technical historians to visually organise and explore entities within the KGIH through accessible and intuitive interfaces without requiring semantic web expertise.

This work [10] presents an approach for supporting curation of cultural heritage knowledge graphs. The work focuses on making knowledge graphs more accessible to human users by addressing barriers associated with navigating and understanding complex graph structures. The proposed approach enables users to curate and organise knowledge graph resources through user-centred interfaces designed for cultural heritage collections. While this work emphasises end-user curation and navigation of cultural heritage knowledge graphs, it does not provide mechanisms for creating persistent thematic collections or narrative groupings of resources. In contrast, the KG Views approach enables historians to organise entities into reusable thematic views that support exploration, interpretation and sharing of historical knowledge.

Sampo-UI [11] is a user interface platform for publishing and exploring KGs through configurable visualisations and application views. It provides a flexible framework for creating domain-specific portals that support browsing, searching and analysing linked data resources. However, the creation and configuration of views within Sampo-UI typically requires technical knowledge of semantic web technologies, data modelling and software configuration, limiting its accessibility for non-technical users. In contrast, the proposed KG Views approach aims to enable non-technical domain experts to create, organise and share meaningful collections of KG resources without requiring expertise in KG technologies.

A KG Editor [12] was previously developed by the VRTI to facilitate the creation, modification and management of resources within the KGIH. While this tool focuses on the direct editing and maintenance of KG content, it does not provide extensive support for the organisation and visualisation of the graph as a whole. The KG Explorer addresses this gap by enabling users to navigate, organise and visualise resources and their relationships within the KG. Building on this functionality, we propose KG Views as a mechanism for grouping resources related to a particular theme, domain, or use case.

While existing approaches provide powerful mechanisms for modelling, curating and exploring KGs, they often require technical expertise or focus primarily on data management and analysis. The proposed KG Views approach addresses this gap by enabling non-technical historians to visually organise, explore

and share thematic collections of KG resources through accessible and intuitive interfaces.

## 7. Future Work and Conclusion

It is planned to further develop the views framework to support a wider range of customisation and interaction capabilities, including improvements to the usability of the editor, enhanced integration of external data sources and additional visualisation features within each view. Future work will also involve further evaluations with historians and other domain experts to better understand how the framework can support different research and exploration workflows. In addition, a key objective is to make the views editor publicly available within the explorer, enabling users to create and manage their own customised views over the KG. Finally, it is planned to extend the approach to support other KGs, which will help evaluate the generalisability of the framework and demonstrate its applicability across a wider range of domains and datasets.

The iterative prototype helped to refine the overall design and functionality of the system through continuous feedback and evaluation. Initial feedback from the historians demonstrated that they were able to effectively use the views editor to facilitate the creation of customised thematic views over the KG. The interface enabled them to organise entities, define relationships between resources and configure metadata without requiring semantic web technical expertise. The feedback also highlighted the usefulness of features such as featured entities, geospatial visualisations and linked views in supporting exploration and discovery within the dataset. As a result, the iterative development process contributed to improving the usability, flexibility and overall effectiveness of the editor for humanities-focused KG exploration.

## Acknowledgments

Virtual Record Treasury of Ireland (VRTI) is funded by the Government of Ireland, through the Department of Tourism, Culture, Arts, Gaeltacht, Sport and Media, under the Project Ireland 2040 framework. The project is also partially supported by the ADAPT Centre for Digital Content Technology under the SFI Research Centres Programme (Grant 13/RC/2106_P2).

## Declaration on Generative AI

During the preparation of this work, the authors used ChatGPT in order to: grammar and spelling check. After using this tool, the authors reviewed and edited the content as needed and takes full responsibility for the publication's content.

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
