# OpenReview forum: "Enabling Non-Technical Users to Create Knowledge Views"
_SEMANTiCS.cc/2026/Workshop/UKG — SEMANTiCS 2026 Workshop UKG Submission_

### Official Review · ~Hannah_Kim6 · 2026-07-18
**The authors developed a user-friendly view editor interface in VRTI-KG explorer platform using a codesign approach, thereby improving non-technical user information management.**

**Rating:** 7
**Confidence:** 3

**Review:**

The authors present a user-friendly view editor for the VRTI-KG Explorer platform that enables historians without prior knowledge of knowledge graphs to manage the visualization of nodes and edges. By involving users throughout the design process, the authors substantially improve the usability and accessibility of the platform. The work addresses an important challenge in making knowledge graph technologies more accessible to non-technical users. However, the manuscript could be strengthened through the following revisions.

* Clarify the use of the term "view." Although the manuscript is generally well written, the term view is used rather loosely. It is reasonable to generate pages from the underlying knowledge graph, but extending the same terminology to subsequent visualizations may be confusing. The authors should provide an operational definition of what constitutes a view.
* Address research ethics for the co-design process. Because the platform was developed through a co-design process involving users, the manuscript should discuss the ethical considerations associated with human participation. Even if the study did not require formal review because users were considered co-designers, the authors should clarify this in the manuscript. Future user studies evaluating the platform's usability or user experience should be conducted under appropriate institutional review board (IRB) approval and follow established ethical guidelines.
* Improve the organization of Section 4 (Use Cases). The paragraph structure in this section is difficult to follow. For example, one paragraph concludes with, "Figure 10 presents the search results page for the VOICES view," while the following paragraph immediately begins by explaining the same figure: "This page indicates to users (A) that they are searching over the entities contained within the VOICES view." Separating the introduction of a figure from its explanation interrupts the flow of the discussion. Reorganizing the section so that each figure is introduced and discussed within the same paragraph would improve readability.
* Reconsider the manuscript organization. In most research articles, the Related Work section appears after the Introduction and before the Methods section. Adopting this more conventional structure would improve the manuscript's organization and align it with common academic writing practices.
* Use a more conventional citation style. The manuscript provides a thorough review of prior work on knowledge graphs in the digital humanities. However, repeatedly referring to previous studies as "This work [9]" or "This work [10]" is unconventional and makes the writing feel repetitive. A more natural style would be to refer to the authors by name or to describe the contribution directly (e.g., "Hyvönen [9] discusses ..." or "Previous work [9] proposes ..."), resulting in smoother and more professional prose.

---

### Official Review · ~Franck_Michel1 · 2026-07-22
**Nice work yet quite technical paper, would benefit from more in-depth analysis and generalisation**

**Rating:** 7
**Confidence:** 4

**Review:**

This short paper presents the concept of KG view implemented to organise and interact with curated list of resources, in the context of the KG of Irish History. An interesting point is the ability for non-technical users to create and manage views.

This is a quite technical paper that reports the status of an on-going development. The writing style is pleasant and clear.
A nice feature is the way graph links are used to provide an intuitive navigation through the graph (between people, views etc.) yet without mentioning or every rendering an actual graph, which may sometimes be counter-productive for non-technical users.

Section 2 about the requirements is extremely short and would deserve a more formal analysis the user’s needs, typically denoting persona, stories, etc.

Section 3 should start with a formal definition of views. Section 3.2 mentions views, sub-views, linked views, but at this point the reader still does not know what a view is. The explanations only come later in section 3.3

The development seems to be ad-hoc, i.e. dedicated specifically to the KGIH. This is quite a concern since it is not meant to be applicable to another KG. Section mentions future plans for generalisation, yet such a question should be considered from the very beginning, as a clear development goal, not at the end when specific code has been written and may be difficult to generalise.
No licence nor pointer to a public repository are mentioned, this lack should be fixed in case of acceptance to promote reuse and adoption.

Typos:
-	“used to associated entities with the view”: “used to associate entities with the view”
-	“This page indicates to users (A) that they are searching over” : misplaced line break before this sentence, should follow the previous sentence

---

### Official Review · ~David_Chaves-Fraga1 · 2026-07-22
**Relevant and Promising Use Case Requiring Further Technical Clarifications**

**Rating:** 7
**Confidence:** 4

**Review:**

This paper presents a use case in which a knowledge graph is made accessible through a web interface in the context of the Virtual Record Treasury of Ireland. The paper is highly relevant to the workshop, as it describes an excellent example of a large number of users benefiting from the information contained in a knowledge graph without needing to be aware of the underlying technology.

The paper’s main contribution is an approach for building “KG Views” on top of the existing knowledge graph, facilitating the exploration of specific subsets of its data. These views are created and managed through a (private?) web interface.

Overall, I find the work interesting and relevant to the workshop. Nevertheless, I have a few minor comments that the authors should consider:

- The requirements section is very brief. It should either be expanded to provide further detail or integrated into another section.
- Why is JSON used as the interchange format? Given the knowledge graph context, would JSON-LD be more appropriate?
- Although the workshop focuses on users interacting with knowledge graphs, it is worth considering whether all the screenshots included in the paper are necessary. A short demonstration video might provide a more effective and engaging presentation of the interface and its workflow.
- It is not entirely clear how users are expected to collect the URIs required to create a KG View without first exploring the underlying graph. Does the interface provide any functionality to help users discover and select the relevant URIs?
- How is a KG View actually materialized from a technical perspective? The paper would benefit from a more detailed explanation of its implementation and underlying representation.
- What would be the cost of generalising this approach to other domains?